# AGENTIC ROBOT: A BRAIN-INSPIRED FRAMEWORK FOR VISION-LANGUAGE-ACTION MODELS IN EMBODIED AGENTS

## ABSTRACT

Long-horizon robotic manipulation poses significant challenges for autonomous systems, requiring extended reasoning, precise execution, and robust error recovery across complex sequential tasks. Current approaches, whether based on static planning or end-to-end visuomotor policies, suffer from error accumulation and lack effective verification mechanisms during execution, limiting their reliability in real-world scenarios. We present Agentic Robot, a brain-inspired framework that addresses these limitations through Standardized Action Procedure (SAP)–a novel coordination protocol governing component interactions throughout manipulation tasks. Drawing inspiration from Standardized Operating Procedures (SOPs) in human organizations, SAP establishes structured workflows for planning, execution, and verification phases. Our architecture comprises three specialized components: (1) a large reasoning model that decomposes high-level instructions into semantically coherent subgoals, (2) a vision-language-action executor that generates continuous control commands from real-time visual inputs, and (3) a temporal verifier that enables autonomous progression and error recovery, ensuring timely subtask termination to avoid redundant execution and enable smooth subgoal transitions. This SAP-driven design supports dynamic self-verification without external supervision. On the LIBERO benchmark, Agentic Robot achieves competitive performance, with a clear advantage in the average success rate of 79.6%, outperforming SpatialVLA by 6.1% and OpenVLA by 7.4% on long-horizon tasks. These results demonstrate that SAP-driven coordination between specialized components enhances both performance and interpretability in sequential manipulation, suggesting significant potential for reliable autonomous systems.

## 1 INTRODUCTION

Recent advances in foundation models have demonstrated remarkable potential for creating embodied agents capable of interpreting natural language instructions and executing complex manipulation tasks (Brohan et al., 2023b; Liang et al., 2023; Kim et al., 2024; Brohan et al., 2023a). These systems effectively bridge the gap between high-level reasoning and low-level physical control. However, existing embodied manipulation systems struggle to achieve reliable performance on long-horizon tasks that require extended sequences of coordinated actions (Huang et al., 2022; Jiang et al., 2022; Feng et al., 2025). Real-world scenarios such as table setting, grocery packing, or furniture assembly demand not only sophisticated reasoning and precise motor control, but also robust error detection and recovery mechanisms throughout extended task execution (Zhu et al., 2021; Chen et al., 2024).

Through extensive analysis of current approaches, we identify fundamental limitations that prevent reliable long-horizon manipulation. Most existing methods fall into two categories with critical weaknesses: static plan-following agents that generate fixed execution sequences without adaptive feedback (Liang et al., 2023; Brohan et al., 2023b), and end-to-end visuomotor policies that directly map observations to actions without intermediate reasoning (Kim et al., 2024). Static planners suffer from compounding error propagation–small deviations early in execution cascade into catastrophic failures (Xu et al., 2022). End-to-end policies lack mechanisms for introspection and often fail to recover from unexpected states, particularly when encountering scenarios outside their training distribution (Zhu et al., 2021).

Drawing insights from Standardized Operating Procedures (SOPs) in human organizations (Hong et al., 2023; Wu et al., 2023), we observe that reliable task execution requires structured coordination protocols. In natural cognition, complex behaviors emerge from specialized neural circuits working through well-defined interaction patterns: prefrontal regions handle planning, motor cortices execute actions, and sensory-motor loops provide continuous verification feedback (Sutton et al., 1999; Nachum et al., 2018). Similarly, in human organizations, SOPs establish clear workflows that minimize errors and enable effective collaboration across different roles. This biological and organizational wisdom suggests that robotic systems can benefit from structured coordination protocols that govern component interactions.

Inspired by these insights, we design **Agentic Robot**, a brain-inspired framework that introduces **Standardized Action Procedure (SAP)**–a novel coordination protocol specifically designed for embodied manipulation tasks. Unlike SOPs, which govern human workflows, SAP encodes the natural cognitive cycle into structured agent interactions for robotic systems. SAP defines the complete agentic loop that governs how our three specialized components–Planner, Executor, and Verifier–coordinate throughout task execution through well-defined interfaces and standardized protocols for information exchange, progress monitoring, and error recovery. Besides, Agentic Robot requires agents to maintain structured interaction protocols throughout the manipulation process. Unlike prior works such as Manipulate-Anything (Duan et al., 2025) that focus on sequential task execution without structured verification, SAP introduces a novel coordination protocol that integrates subgoal-level verification and proactive recovery. More specifically, all components follow strict SAP-defined workflows, ensuring that information handoffs comply with established protocols and eliminating the communication breakdowns that plague existing systems.

**Our main contributions are as follows:**

- We introduce Agentic Robot, a brain-inspired agentic framework for embodied manipulation that incorporates structured coordination protocols. The framework is highly modular and interpretable, with well-defined component interfaces, making it a powerful platform for developing reliable long-horizon manipulation systems.

- We propose Standardized Action Procedure (SAP), a novel coordination protocol that governs the complete agentic loop in robotic manipulation tasks. SAP encodes structured interactions between planning, execution, and verification phases, enhancing system reliability and reducing error propagation through standardized workflow management.

- We achieve competitive performance on the LIBERO benchmark with an average success rate of 79.6%. Extensive experimental results convincingly demonstrate that our SAP-driven approach represents a promising framework for reliable embodied manipulation, with particularly strong improvements on challenging long-horizon tasks.

## 2 AGENTIC ROBOT FRAMEWORK: A BRAIN-INSPIRED CONTROL LOOP

### 2.1 OVERVIEW

We introduce Agentic Robot, an agentic framework that reformulates long-horizon manipulation as a closed perception-reasoning-execution-verification loop, inspired by biological cognition and multi-agent LLM systems (Hong et al., 2023; Wu et al., 2023). Drawing insights from SOPs that govern effective human workflows, we propose SAP–a novel coordination protocol that structures component interactions throughout the manipulation process. SAP establishes explicit protocols for information exchange, progress monitoring, and error recovery, enabling robust execution of complex manipulation tasks. Our design is grounded in recent advances across large reasoning models (LRMs), vision-language models (VLMs), and vision-language-action (VLA) systems. We provide a detailed review of these foundations in the Related Works section (see Appendix B).

Our architecture integrates three specialized components: (1) a planner based on LRM that decomposes high-level instructions into structured subgoals, (2) an executor based on VLA that generates continuous control actions from subgoals and visual input, and (3) a verifier based on VLM that conducts self-assessment for autonomous progression or recovery. Each component operates within the SAP framework, following standardized interfaces and communication protocols that ensure seamless coordination throughout task execution.

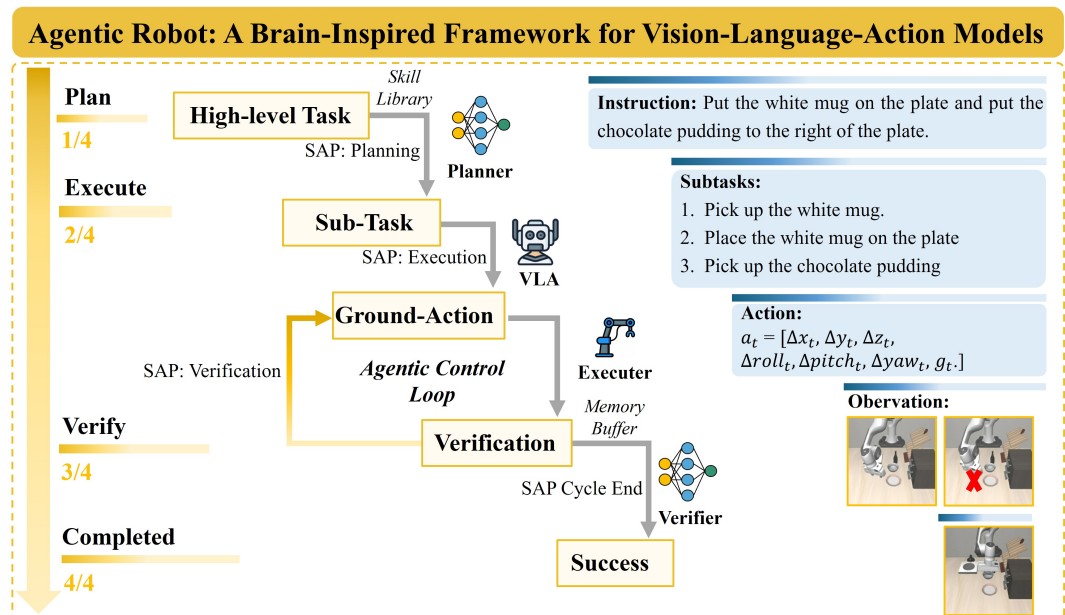

Figure 1: Overview of the Agentic Robot framework governed by Standardized Action Procedure (SAP). (1) A high-level task is decomposed into structured subgoals by an LRM-based planner, guided by a skill library. (2) A VLA policy as Executor executes each subgoal using natural language instructions and real-time visual input. (3) A VLM-based verifier periodically inspects a sliding window of third-person and wrist-mounted views to determine whether to continue, retry, or recover. This SAP-driven agentic loop enables robust, interpretable, and feedback-driven manipulation.

As shown in Fig. 1, our agent processes task descriptions and RGB observations from third-person and egocentric cameras. The planner generates subgoals following SAP specifications, which the VLA model translates into 7-DoF actions based on visual input. Simultaneously, the verifier monitors a temporal frame buffer to determine subgoal completion according to SAP verification protocols, moving to the next subgoal upon success, thus not only preventing redundant re-execution but also acting as the switch that governs subgoal-to-subgoal transitions within the SAP cycle. This architecture implements a sequence of agentic steps, each combining intention grounding, visuomotor execution, and perception-based validation within the SAP framework, enabling execution correction without external supervision.

## 2.2 PLANNER: LRM FOR SUBGOAL GENERATION

The planner module, denoted as $P$, functions as the high-level reasoning component within our SAP framework. It converts task instructions $T$ into a structured sequence of executable subgoals following standardized decomposition protocols:

$$\{t_1, t_2, \ldots, t_N\} = P(T, I_0), \tag{1}$$

where $I_0$ represents the initial visual observation. Each subgoal $t_i$ forms a complete and constrained instruction derived from an Atomic Skill Library (Li et al., 2025), which defines standardized action templates such as:

```
pick up [object] — place [object] in/on [location] — turn on/off
                          [device]
```

This constrained approach ensures compatibility with the executor while maintaining interpretability of the execution pipeline, adhering to SAP principles of structured component interaction.

We implement the planner using a state-of-the-art large multimodal reasoning model (e.g., GPT-4o), which processes both the instruction $T$ and optionally an image $I_0$ for visual grounding. The SAP-compliant prompt architecture includes three structured components: (1) a task preamble explaining the planner's role within the framework, (2) the complete Atomic Skill Library specifying

allowed action types, and (3) carefully selected few-shot examples demonstrating proper subgoal decomposition. These examples guide the model to establish appropriate task boundaries, resolve ambiguities, and break down complex instructions into 2-5 atomic steps. Through extensive validation, we determined that subgoals with 1-2 semantic units (e.g., verb + object or verb + object + location) achieve optimal balance between clarity and executability within the SAP framework.

## 2.3 VLA EXECUTOR: REACTIVE VISUOMOTOR POLICY

The executor module $E$ serves as the core visuomotor interface that transforms each subgoal $t_i$ and associated visual observations $I_t^r$ into continuous low-level control signals $\mathbf{a}_t$ according to SAP execution protocols:

$$\mathbf{a}_t = \pi_{\text{exec}}(t_i, I_t^r), \tag{2}$$

where $\mathbf{a}_t \in \mathbb{R}^7$ represents the robot's Cartesian displacement and gripper configuration. The first six dimensions encode translation and rotation vectors, while the last component $g_t \in \{0, 1\}$ indicates the binary gripper state.

We utilize OpenVLA (Kim et al., 2024), an open-source pretrained VLA model that establishes direct connections between natural language subgoals and visual observations. Each subgoal adheres to the structured format outlined in our Atomic Skill Library, enabling the VLA model to systematically generate actions by understanding both language instructions and visual scene content. This structured approach enhances compatibility and interpretability across manipulation scenarios while constraining the action space to physically feasible trajectories.

Despite its stateless design, the executor integrates robust error-handling capabilities through the SAP verification loop. When execution failures occur, the standardized verification mechanism detects issues through visual assessment and triggers specific recovery actions following SAP protocols. If multiple recovery attempts fail, the framework marks the task as failed and halts execution to prevent unsafe behaviors. This closed-loop error detection significantly improves system robustness–particularly in long-horizon tasks–by reducing cascading errors. As shown in Section H, our method achieves up to 24% improvement over OpenVLA on the Bowl-Drawer task and 21% on Soup-Sauce, confirming the effectiveness of subgoal-level verification and recovery.

## 2.4 VERIFIER: PERCEPTION-BASED SUBGOAL ASSESSMENT AND RECOVERY

The verifier module $V$ provides critical feedback within the SAP framework by assessing the success of each subgoal $t_i$ through visual analysis. For every verification step $t_v$, it produces a binary response following a two-stage assessment protocol:

$$\hat{y}_{t_v} = \pi_{\text{ver}}(\mathcal{B}_{t_v}, t_i) \rightarrow \texttt{Yes} \text{ or } \texttt{No}, \tag{3}$$

where $\mathcal{B}_{t_v} = \{(I_{t_v-k}^r, I_{t_v-k}^w)\}_{k=0}^{K-1}$ is a sliding buffer of recent image pairs from third-person and wrist-mounted views. This temporal buffer captures visual dynamics such as object displacement or contact transitions, typically with $K = 2$ and frame intervals of 5. This explicit success detection serves two critical purposes: (i) it prevents repetitive execution of already accomplished subtasks, and (ii) it provides a reliable gating signal for transitioning to the next subgoal in the task sequence.

We employ Qwen2.5-VL-3B-Instruct (Bai et al., 2025) as the verifier model to evaluate whether subgoal $t_i$ is complete. The verification prompt follows SAP's structured format: "Based on the image sequence, has the robot successfully completed [subgoal]?" The model is fine-tuned with LoRA (Hu et al., 2022) on a dataset of annotated triplets $(\mathcal{B}_t, t_i, y)$ with $y \in \{\texttt{Yes}, \texttt{No}\}$. To adapt the verifier to subgoal-level introspection, we fine-tune Qwen2.5-VL-3B-Instruct using LoRA on a compact dataset of approximately 500 annotated triplets. Despite its small scale, the dataset covers diverse subgoal types and visual scenes, and leverages strongly structured prompts to guide learning. This setting demonstrates that even with limited supervision, targeted adaptation can yield effective visual verification in closed-loop execution (see Section 3.3).

When the initial response is $\hat{y}_{t_v} = \texttt{No}$, the verifier performs a secondary check to determine whether the robot is stuck:

$$f_t = \pi_{\text{diag}}(\mathcal{B}_{t_v}) \rightarrow \texttt{Stuck} \text{ or } \texttt{StillTrying}, \tag{4}$$

where $\pi_{\text{diag}}$ is a diagnosis module that detects conditions such as stationary arms, failed grasp, or oscillating behaviors. If $f_t = \texttt{Stuck}$, a recovery action is triggered:

$$\mathbf{a}_{t+1} = \pi_{\text{rec}}(f_t, O_{t+1}), \tag{5}$$

such as lifting the gripper or reorienting the wrist. The system then re-executes $t_i$ and resumes the same two-stage verification process at the next interval. After $R_{\max}$ unsuccessful recovery attempts, the task is marked as failed.

To optimize responsiveness and efficiency, verification is performed every 20 frames (i.e., $f_{\text{ver}} = 0.5\,\text{Hz}$), achieving near-optimal accuracy (only 1.2% drop from 10-frame intervals) while reducing computational load by 48%. Compared to single-pass goal-checking methods, our two-level verifier allows for mid-execution correction and fine-grained failure localization.

## 2.5 SAP: Standardized Action Procedure for Coordinated Agentic Control

In robotic manipulation, the absence of structured coordination protocols often leads to execution failures, particularly in long-horizon tasks where accumulated errors and lack of systematic verification result in task breakdown. Drawing inspiration from Standardized Operating Procedures (SOPs) that have proven effective in complex collaborative environments, we introduce **Standardized Action Procedure (SAP)** as a systematic framework that encodes proven coordination patterns into robotic agentic systems.

SAP represents a principled approach to orchestrating closed-loop execution within our agentic robot framework by establishing *standardized coordination protocols* across perception, planning, execution, and verification components. The core design philosophy rests on three fundamental principles: (1) **Modular Decomposition** - complex manipulation tasks are systematically decomposed into manageable, verifiable subgoals as illustrated in Fig. 2; (2) **Structured Coordination** - component interactions follow predefined workflows rather than opportunistic communication; and (3) **Adaptive Verification** - systematic checkpoints enable early error detection and recovery.

### 2.5.1 SAP Operational Framework

Each SAP cycle at time $t$ constitutes an *agentic step* that encapsulates the complete perception-planning-execution-verification workflow:

$$\mathcal{S}_t = (O_t, t_i, \mathbf{a}_t, \hat{y}_t), \qquad (6)$$

where $O_t = \{I_t^r, I_t^w\}$ denotes egocentric and third-person views, $t_i$ represents the current subgoal within the structured task decomposition, $\mathbf{a}_t$ is the executed action, and $\hat{y}_t \in \{\text{Yes}, \text{No}\}$ indicates the verification result.

SAP defines four specialized components with standardized interfaces and coordination protocols:

**(1) Multimodal Perception.** At each time step, the agent collects dual-perspective observations:

$$O_t = \{I_t^r, I_t^w\} \in \mathcal{I}_r \times \mathcal{I}_w, \qquad (7)$$

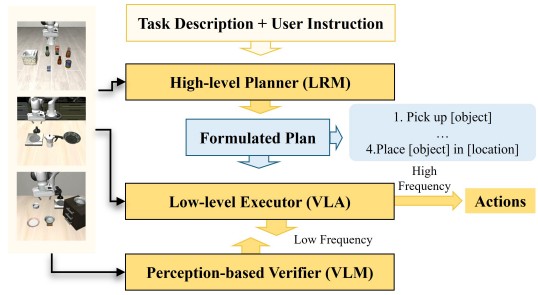

Figure 2: SAP flow. The LRM planner converts instructions into structured subgoals using a skill library, which are then executed by a VLA executor and verified by a VLM verifier.

which provides comprehensive workspace understanding following standardized observation protocols.

**(2) Formulated Plan.** The planner $P$ converts task instructions $T$ into a structured sequence of executable subgoals following standardized decomposition protocols:

$$\{t_1, t_2, \ldots, t_N\} = P(T, I_0), \qquad (8)$$

where $I_0$ represents the initial visual observation and each subgoal $t_i$ is derived from the Atomic Skill Library.

**(3) Reactive Execution.** The executor translates subgoal $t_i$ into low-level control signals:

$$\mathbf{a}_t = \pi_{\text{exec}}(t_i, O_t), \qquad (9)$$

where $\pi_{\text{exec}}$ maps semantic goals and current vision into 7-DoF actions following standardized execution protocols.

**(4) Temporal Verification.** Every $\Delta t_v$ frames (typically 20), the verifier performs systematic evaluation:

$$\hat{y}_{t_v} = \pi_{\text{ver}}(\mathcal{B}_{t_v}, t_i), \quad f_t = \pi_{\text{diag}}(\mathcal{B}_{t_v}), \tag{10}$$

where $\mathcal{B}_{t_v} = \{(I_t^r, I_t^w)\}_{k=0}^{K-1}$ represents the sliding buffer of recent image pairs. If $\hat{y}_{t_v} = \text{Yes}$, the agent proceeds to the next subgoal. If not, and $f_t = \text{Stuck}$, a recovery action is triggered:

$$\mathbf{a}_{t+1} = \pi_{\text{rec}}(f_t, O_{t+1}). \tag{11}$$

SAP execution is managed by an asynchronous finite-state machine $\mathcal{M}_{\text{SAP}}$ with component-specific frequencies: the executor operates at 10 Hz ($\Delta t_{\text{exec}} = 0.1\text{s}$), and the verifier at 0.5 Hz ($\Delta t_{\text{ver}} = 2\text{s}$). By enforcing structured control cycles with modular boundaries and layered feedback, SAP enhances agent reliability and interpretability. It supports in-situ correction, isolates errors, and ensures safe recovery, addressing core limitations of open-loop or end-to-end systems in dynamic and uncertain manipulation environments.

## 3 EXPERIMENTS

### 3.1 EXPERIMENTAL SETUP

We evaluate our Agentic Robot framework on long-horizon manipulation tasks in simulated embodied environments. The agent employs a dual-camera perception system: a static agent-view camera for global scene context and a wrist-mounted eye-in-hand camera for local detail. Both cameras provide synchronized RGB observations at each timestep. The action space consists of a 7-dimensional continuous control vector representing 6-DoF end-effector control plus a binary gripper state.

**Benchmarks.** We carry out evaluations using the LIBERO benchmark suite (Liu et al., 2023), which provides a standardized way to assess instruction-following manipulation across various environments. Our experiments concentrate on four specific challenge subsets: LIBERO-Spatial, which focuses on understanding spatial relationships; LIBERO-Object, which tests generalization to new objects; LIBERO-Goal, which assesses abstract goal execution; and LIBERO-Long, which involves extended sequential manipulations. Each subset consists of 10 distinct tasks, and for each task, there are 50 human-teleoperated demonstrations.

**Baselines.** We benchmark our approach against the following generalist policies, including previous state-of-the-art open-sourced models: Diffusion Policy (Chi et al., 2023), Octo-Base (Team et al., 2024), OpenVLA (Kim et al., 2024), TraceVLA Zheng et al. (2024), and SpatialVLA (Qu et al., 2025). These methods represent a variety of model paradigms, including diffusion-based control, transformer-based visuomotor policies, and large-scale vision-language-action models. For fair comparison, we follow the original hyperparameters and evaluation settings as reported in their respective works without additional tuning. Detailed information are provided in Appendix A.2.

**Implementation.** Agentic Robot integrates the three modules described in Section 2: a GPT-4o-based planner for subgoal decomposition, an OpenVLA-based executor for visuomotor control, and a fine-tuned Qwen2.5-VL-3B-Instruct verifier for subgoal completion assessment. For error recovery, we raise the gripper to a safe position upon failure detection before re-evaluation. Unless otherwise specified, verification is performed every 20 frames.

### 3.2 MAIN RESULTS

Table 1 presents success rates across four LIBERO benchmark suites. Agentic Robot achieves competitive performance with 79.6% average success rate, surpassing all baselines across diverse manipulation scenarios.

**Cross-domain generalization.** Agentic Robot consistently ranks among the top-3 performers across all task categories, demonstrating exceptional versatility in diverse manipulation scenarios. Unlike specialized approaches that exhibit domain-specific excellence but inconsistent cross-domain performance, our framework maintains balanced efficacy throughout the benchmark suite. This strong generalization indicates that our architecture effectively captures essential manipulation principles that go beyond task-specific requirements, which is a critical capability for deployment in unconstrained real-world environments.

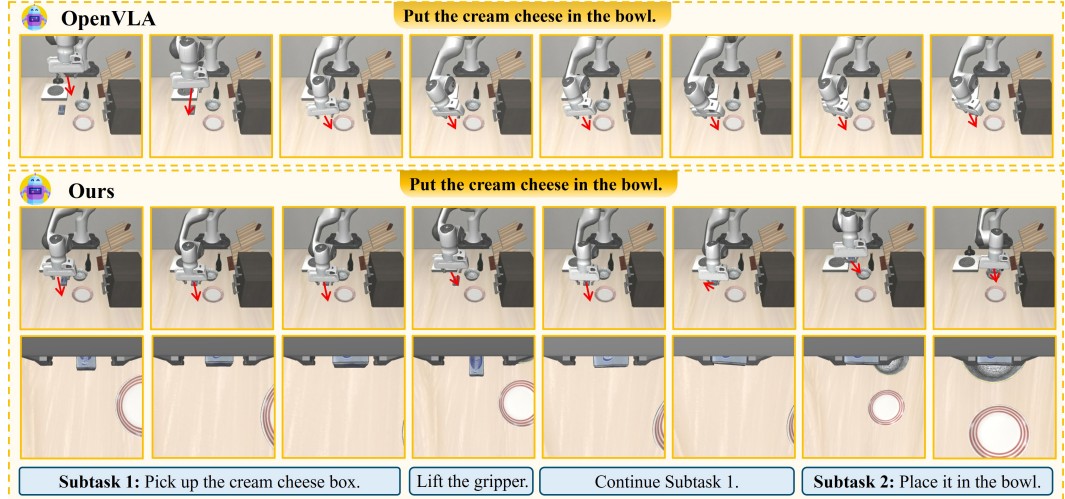

Figure 3: Comparison between OpenVLA and Agentic Robot on the task "Put the cream cheese in the bowl." **Top:** OpenVLA fails to grasp the object, causing the gripper to collide with the table and the task to fail. **Bottom:** Agentic Robot decomposes the task into subgoals and detects failure via visual verification. It issues a recovery action (`Lift the gripper`) and completes the task through retry.

Table 1: LIBERO benchmark results comparing success rates (SR, %) with rank (in parentheses) across four task suites, averaged over three random seeds with 500 evaluation trials. FT indicates fine-tuning on task-specific demonstrations. Bold values highlight the best performance.

| Method | LIBERO-Spatial SR (Rank) | LIBERO-Object SR (Rank) | LIBERO-Goal SR (Rank) | LIBERO-Long SR (Rank) | Average SR (Rank) |
|---|---|---|---|---|---|
| Diffusion Policy (Chi et al., 2023) | 78.3 ± 1.1 (6) | **92.5 ± 0.7** (1) | 68.3 ± 1.2 (6) | 50.5 ± 1.3 (6) | 72.4 ± 0.7 (6) |
| Octo-Base (FT) (Team et al., 2024) | 78.9 ± 1.0 (5) | 85.7 ± 0.9 (5) | **84.6 ± 0.9** (1) | 51.1 ± 1.3 (5) | 75.1 ± 0.6 (4) |
| OpenVLA (FT) (Kim et al., 2024) | 84.7 ± 0.9 (3) | 88.4 ± 0.8 (4) | 79.2 ± 1.0 (3) | 53.7 ± 1.3 (4) | 76.5 ± 0.7 (3) |
| TraceVLA (FT) (Zheng et al., 2024) | 84.6 ± 0.2 (4) | 85.2 ± 0.4 (6) | 75.1 ± 0.3 (5) | 54.1 ± 1.0 (3) | 74.8 ± 0.5 (5) |
| SpatialVLA (FT) (Qu et al., 2025) | **88.2 ± 0.5** (1) | 89.9 ± 0.7 (2) | 78.6 ± 0.6 (4) | 55.5 ± 1.0 (2) | 78.1 ± 0.7 (2) |
| **Agentic Robot (Ours)** | 85.8 ± 0.6 (2) | 89.0 ± 0.8 (3) | 81.8 ± 0.8 (2) | **61.6 ± 1.2** (1) | **79.6 ± 0.8** (1) |

**Long-horizon planning.** In the particularly challenging LIBERO-Long tasks, Agentic Robot significantly outperforms all baseline methods, achieving a 6.1% improvement over SpatialVLA, the previous state-of-the-art. This substantial enhancement is a direct result of our core architectural innovation: the breakdown of complex instructions into individually verifiable subgoals with clearly defined intermediate checkpoints. By implementing closed-loop verification at the subgoal level, our system effectively reduces error accumulation, which is a major limitation of existing approaches, especially as task complexity increases. The relationship between task horizon length and performance advantage is further examined in Section H, where we demonstrate that Agentic Robot's performance advantage grows in proportion to task complexity.

**Verification and recovery.** A key innovation in Agentic Robot is its explicit subgoal-level verification and recovery mechanism. Unlike end-to-end baselines relying on implicit success estimation, our system provides transparent execution monitoring with interpretable assessments, enabling targeted recovery strategies. This retry mechanism enhances performance across all benchmarks and proves particularly effective in complex multi-step tasks, which is illustrated in Fig. 3. Without recovery, baselines such as OpenVLA continue executing the current subgoal indefinitely, even when the gripper is clearly stuck, for instance, when it presses against the table after a failed grasp. In contrast, Agentic Robot uses visual verification to detect such failure states and triggers a simple recovery behavior: lifting the gripper vertically before retrying the action. As shown in the figure, this enables the robot to reattempt grasping and proceed to the next subgoal successfully. Although our current recovery policy is deliberately minimal and does not resolve all failure cases, it demonstrates the

potential of incorporating visual feedback loops for execution robustness. Future extensions may incorporate more sophisticated recovery strategies, such as policy rollback, re-grasping, or online subgoal regeneration, to further enhance success rates under real-world uncertainties.

### 3.3 ABLATION STUDY

To analyze each component's contribution, we conduct a systematic ablation study on LIBERO-Long, where extended manipulation sequences are particularly sensitive to architectural modifications. Table 2 summarizes our findings.

**Multimodal Planning.** When the planner is restricted to text-only inputs, the task success rate drops to 57.4%, representing a 4.2% decline relative to the full multimodal setting. This performance gap highlights the importance of visual context for grounding instructions and resolving object ambiguity, particularly in cluttered scenes.

**Recovery Mechanism.** Removing the recovery routine after failed subgoal verification leads to a reduced success rate of 59.7%, reflecting a 1.9% degradation. This result confirms the value of even minimal corrective behaviors in mitigating error accumulation across long-horizon task sequences.

**Verification Quality.** Substituting the fine-tuned verifier with a zero-shot VLM yields a substantial drop in success rate to 35.3%, indicating a degradation of 26.3%. This sharp decline suggests that generic models are insufficiently sensitive to subtle changes in scene state, and that domain adaptation is essential for accurate subgoal-level introspection. As illustrated in Fig. 6, the zero-shot verifier fails to detect subgoal completion, causing repetitive execution and eventual failure. In contrast, our model enables sequential verification and completion of each subtask. Quantitatively, it improves the LIBERO-Long success rate by 26.3% over the zero-shot baseline, confirming the value of even small-scale domain adaptation.

Table 2: Ablation study on LIBERO-Long. Each row represents the success rate (SR) after removing a key component.

| Setting | SR (%) |
|---|---|
| No Visual Input | 57.4 |
| No Recovery Mechanism | 59.7 |
| No Fine-tuned VLM | 35.3 |
| No Subgoal Decomposition | 53.7 |
| **Full System** | **61.6** |

**Hierarchical structure.** Without subgoal decomposition (i.e., vanilla OpenVLA), requiring the system to perform instructions as single, monolithic goals results in a performance drop to 53.7%, a decrease of 7.9%. This supports our main hypothesis that breaking complex tasks into smaller, atomic skills significantly enhances execution reliability and verification accuracy. As illustrated in Fig. 4, the lack of subgoal decomposition causes the executor to attempt the full task without the ability to recover from intermediate failures, ultimately skipping necessary steps and leading to task failure. In contrast, our hierarchical agent sequentially verifies and completes each subtask, ensuring successful full-task completion. Each component offers measurable benefits, with fine-tuned verification and hierarchical planning contributing the most substantial improvements. These findings confirm the effectiveness of our Agentic Robot for handling long-horizon manipulation tasks.

## 4 DISCUSSION AND LIMITATIONS

We now reflect on the design of Agentic Robot, highlighting its strengths and pointing out remaining challenges.

**Verification as a robustness mechanism.** A central contribution of our framework is the introduction of visual verification as a control signal for subgoal progression. The verifier functions as a semantic gatekeeper that determines whether to proceed, retry, or terminate, enabling subgoal-level error detection and correction without access to ground-truth state information. Our empirical results demonstrate this approach's effectiveness in mitigating compounding errors, particularly in long-horizon settings where early mistakes could cascade through subsequent action sequences. The incorporation of recovery behaviors further enhances system resilience under environmental uncertainty and partial observability. Beyond robustness, verification also functions as a termination criterion, ensuring the agent exits completed subtasks promptly and transitions to the next goal without unnecessary repetition.

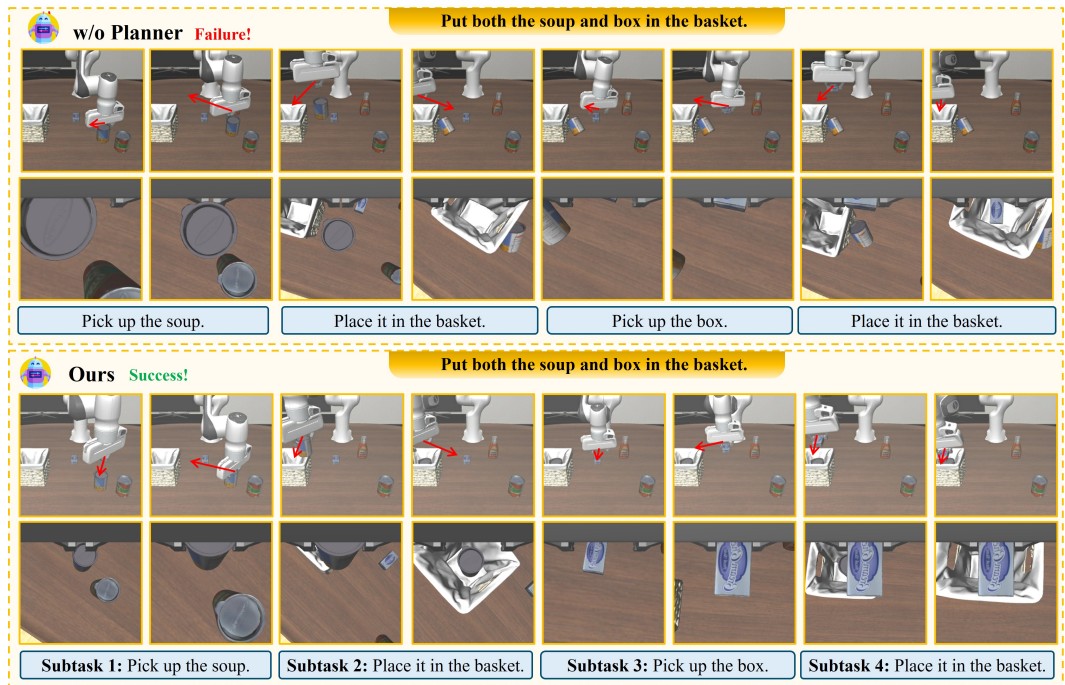

Figure 4: Comparison between **w/o Subgoal Decomposition** and **Ours (w/ Subgoal Decomposition)** on the task: "Put both the soup and the box in the basket." **Top:** Without subgoal decomposition, the entire instruction is passed directly to the executor, which attempts to complete the full task in one shot. However, failure in an intermediate subtask (e.g., placing the soup in the basket) is not detected or recovered, leading to skipped actions and overall task failure. **Bottom:** With explicit subgoal decomposition, the planner breaks down the instruction into sequential subtasks, each verified by the VLM before proceeding. This enables step-by-step execution and recovery, resulting in the successful completion of the full task.

**Real-World deployment challenges.** While our results are validated in high-fidelity simulated environments, transferring Agentic Robot to physical platforms introduces several challenges. These include handling sensor noise in RGB inputs, adapting to real-world lighting variations and occlusions, and compensating for actuation delays. Furthermore, the verifier's robustness to visual domain shift requires extensive evaluation. Future work will incorporate domain adaptation and sim-to-real transfer techniques, particularly focusing on real-image fine-tuning for both the verifier and executor components to mitigate these challenges.

## 5 CONCLUSION

This work introduces Agentic Robot, a brain-inspired framework that uses Standardized Action Procedure (SAP) to improve reliability and interpretability in robotic manipulation systems. The framework decomposes complex tasks into coordinated interactions between three specialized components– planner, executor, and verifier–operating through well-defined SAP protocols that mirror biological cognition. By establishing explicit protocols for component communication, progress monitoring, and failure recovery, SAP addresses fundamental limitations in existing manipulation systems while allowing independent component optimization through standardized interfaces. Extensive validation on the LIBERO benchmark demonstrates competitive performance with 79.6% average success rate, including substantial improvements of 24% on Bowl-Drawer tasks and 21% on Soup-Sauce tasks. This successful integration of brain-inspired architectures with standardized coordination protocols shows how biologically motivated design principles can enhance both performance and interpretability in embodied AI systems.

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

*Appendix of paper "Agentic Robot: A Brain-Inspired Framework for Vision-Language-Action Models in Embodied Agents"*

## DECLARATION OF LLM USAGE

We used large language models (LLMs) as assistive tools in the preparation of this paper. Specifically, LLMs were employed for language editing and improving clarity. All research ideas, methodologies, theoretical results, and experiments were conceived and conducted by the authors. The authors take full responsibility for the content of this paper.

## A  DETAILED EXPERIMENT SETUP

### A.1  BENCHMARK

Our primary evaluation is conducted on the **LIBERO** suite, a benchmark collection for instruction-following manipulation in diverse simulated environments. We select the four suites **LIBERO-Long**, **LIBERO-Spatial**, **LIBERO-Object**, and **LIBERO-Goal** each comprising 10 tasks and 50 human-teleoperated demonstrations per task. The multitask performance of the pretrained VLA policy is evaluated on these suites. Specifically:

- **LIBERO-Spatial**: Contains the same set of objects but in varying layouts, testing the model's ability to understand spatial relationships. Example language instruction: *pick up the black bowl in the top drawer of the wooden cabinet and place it on the plate.*
- **LIBERO-Object**: Features consistent scene layouts but introduces different objects, evaluating the model's understanding of object types. Example language instruction: *pick up the chocolate pudding and place it in the basket.*
- **LIBERO-Goal**: Maintains the same objects and layouts while varying task goals, assessing the model's knowledge of diverse task-oriented behaviors. Example language instruction: *turn on the stove.*
- **LIBERO-Long** (also referred to as **LIBERO-10**): Comprises long-horizon tasks involving diverse objects, layouts, and task goals, challenging the model's ability to handle extended planning and execution. Example language instruction: *pick up the book and place it in the back compartment of the caddy.*

### A.2  BASELINES

**Baselines.** We benchmark our approach against the following generalist policies, including previous state-of-the-art open-sourced models:

- **Diffusion Policy (Chi et al., 2023)**: A way of generating robot behavior by representing a robot's visuomotor policy as a conditional denoising diffusion process.
- **Octo-Base (Team et al., 2024)**: A 93M parameter transformer-based policy trained on 800k trajectories from the Open-X-Embodiment (O'Neill et al., 2024) Dataset.
- **OpenVLA (Kim et al., 2024)**: A 7B parameter VLA trained on the Open-X-Embodiment (O'Neill et al., 2024) Dataset, representing large-scale generalist policies.
- **TraceVLA (Zheng et al., 2024)**: Finetuned from OpenVLA with visual trace prompting.
- **SpatialVLA (Qu et al., 2025)**: A 4B parameter VLA trained on 1.1 million real-world robot episodes.

## B  RELATED WORKS

### B.1  LARGE REASONING MODELS

Recent progress in large reasoning models (LRMs) has dramatically improved general-purpose cognitive capabilities, providing a foundation for downstream embodied agents (Tie et al., 2025).

Models such as `DeepSeek-R1` (Guo et al., 2025) adapt rule-based reinforcement learning pipelines into massive 671B-parameter text models, enabling distilled checkpoints for multimodal adaptation. `Gemini-2.5` Team et al. (2023) unifies vision, audio, and long-text inputs into a state-of-the-art retrieval system over 10M-token contexts. `MM-Eureka` (Meng et al., 2025) introduces visual-math chain-of-thought training with reinforcement learning, achieving new benchmarks on multimodal math reasoning. Similarly, `VLM-R1` (Shen et al., 2025) transfers the R1 architecture into the vision-language domain, yielding strong zero-shot visual reasoning capabilities. Although these models offer unprecedented perceptual and reasoning skills, they are not explicitly trained for robotics tasks, leaving open the challenge of grounding abstract reasoning into actionable physical plans, particularly in long-horizon settings.

## B.2 VISION-LANGUAGE MODELS IN ROBOTICS

Vision-language models (VLMs) have increasingly been adapted into robotic systems to bridge perception and semantic understanding (Zhang et al., 2025b). `PaLM-E` (Driess et al., 2023) integrates vision tokens into a 562B parameter LLM, enabling long-horizon real-world manipulation while retaining general VQA abilities. `VoxPoser` (Huang et al., 2023) leverages CLIP and LLM prompting to synthesize volumetric value maps for zero-shot pick-and-place. In navigation, `VLM-Social-Nav` (Song et al., 2024) grades candidate trajectories through captioning models to enhance social compliance, while `NaVid` (Zhang et al., 2024) predicts step-wise actions from videos and language for map-free instruction following. `GSON` (Luo et al., 2024) further extends visual reasoning to group-aware navigation by detecting social formations. While these systems demonstrate the semantic reasoning power of VLMs, they primarily focus on static goal conditions or trajectory scoring, rather than dynamically verifying subgoal progress within a continuous manipulation task as required by long-horizon execution.

## B.3 VISION-LANGUAGE-ACTION MODELS

Vision-Language-Action (VLA) models directly map visual observations and language instructions to robotic control actions (Yue et al., 2024; Liu et al., 2024; Wang et al., 2024). Notable examples include `RT-2` (Zitkovich et al., 2023), which co-trains a VLM with robot episodes using action tokenization, and `OpenVLA` (Kim et al., 2024), which scales this approach with 970k demonstrations, outperforming `RT-2-XL` while maintaining efficiency. Advances like `ChatVLA` (Zhou et al., 2025) incorporate phased alignment and policy routing to preserve reasoning capabilities during execution, while `ECoT` (Zawalski et al., 2024) introduces chain-of-thought reasoning for improved performance on long-horizon tasks. `RoboMM` (Yan et al., 2024) achieves cross-domain generalization through modality-isolation masking, `SpatialVLA` (Qu et al., 2025) enhances 3D spatial understanding, and `TraceVLA` (Zheng et al., 2024) improves spatio-temporal awareness by visualizing robot trajectory traces. Recent extensions further broaden the design space. `CoPa` (Huang et al., 2024) grounds object parts into spatial constraints for fine-grained manipulation, `Manipulate-Anything` (Duan et al., 2025) automates large-scale demonstration generation with VLM-based reasoning, `RoBridge` (Zhang et al., 2025a) introduces a hierarchical cognition–execution bridge, and `ReKep` (Huang et al., 2025) formulates relational keypoint constraints for multi-stage and bimanual tasks. Despite their diverse strategies, these systems lack a formalized coordination protocol, omit explicit verifiers for systematic subgoal monitoring, and do not adopt modular SAP-style loops. Thus, although VLA models represent a step toward end-to-end instruction following, they generally overlook mechanisms for modular coordination, explicit verification, and recovery protocols, which are crucial for robust execution in complex long-horizon environments.

## C ATOM SKILL LIBRARY

An atomic skill library is a dynamically expanding repository of fine-grained manipulation skills that a robot can invoke to perform complex end-to-end tasks without retraining a monolithic policy (Li et al., 2025). In our framework, the atomic skill library is a curated collection of low-level manipulation primitives, such as "pick up [object]," that have been manually defined by domain experts to guarantee predictable performance and semantic clarity. Each atomic skill encapsulates a self-contained control policy or trajectory generator, allowing complex tasks to be decomposed into a sequence of reusable, verifiable subroutines. Although hand-crafting these primitives ensures reliability and interpretability,

the same library structure can be populated or even expanded automatically by large language models: an LLM can translate high-level task descriptions into candidate skill definitions or suggest refinements to existing primitives. By combining expert-driven design with LLM-powered generation, this hybrid approach accelerates skill coverage, simplifies adaptation to novel tasks, and preserves the modularity and transparency crucial for robust robotic operation. The atom skill library is detailed as follows:

```
pick up [object] (from [location]/[object])
 place [object] in/on [location]/[object]
         push [object] to [location]
place [object] to the [direction] of [object]
   open/close [object/container/drawer/etc.]
               turn on/off [device]
```

## D EXPERIMENTS ON TASK DIVISION

### D.1 DESCRIPTION OF LIBERO-LONG

Now, we give the detailed task description of LIBERO-Long as used in Section H.

- **Soup-Sauce:** The robot must locate and pick up both the alphabet soup can and the tomato sauce can. It then needs to place both items inside a basket. This task tests multi-object handling and proper placement.

- **Cheese-Butter:** In this task, the agent must pick up two items: a box of cream cheese and a piece of butter. Both need to be placed into the same basket. The challenge involves identifying similar-looking food items and executing sequential pick-and-place actions.

- **Stove-Moka:** The robot is required to first turn on a stove, then place a moka pot on top of it. This involves both environment interaction (activating the stove) and precise object placement. It tests sequential decision-making and tool use.

- **Bowl-Drawer:** The task requires the agent to open a bottom drawer of a kitchen cabinet, place a black bowl inside, and then close the drawer. This combines manipulation of articulated components (the drawer) with careful object handling.

- **Mug-Mug:** The agent needs to distinguish two mugs by color and place them on specific plates: the white mug on the left plate, and the yellow-and-white mug on the right. The task emphasizes color-based object recognition and spatial arrangement.

- **Book-Caddy:** The robot must pick up a book and place it into the rear compartment of a caddy organizer. This task involves handling flat objects and placing them into confined spaces, testing precision and spatial reasoning.

- **Mug-Pudding:** The robot places a white mug onto a plate and then positions a chocolate pudding to the right of that plate. It requires understanding relative spatial positioning between objects and accurate placement.

- **Soup-Cheese:** This task is similar to task 1 but with a different object combination: alphabet soup and cream cheese box. The agent must place both items into a basket, reinforcing generalization across similar multi-object tasks.

- **Moka-Moka:** The robot needs to find two moka pots and place both on the stove. The challenge lies in handling duplicate objects and placing them correctly on the same surface.

- **Mug-Wave:** In this task, the agent must place a yellow-and-white mug inside a microwave and then close the microwave door. It tests interaction with articulated appliances and precise object insertion.

## D.2 Illustration of Task Division

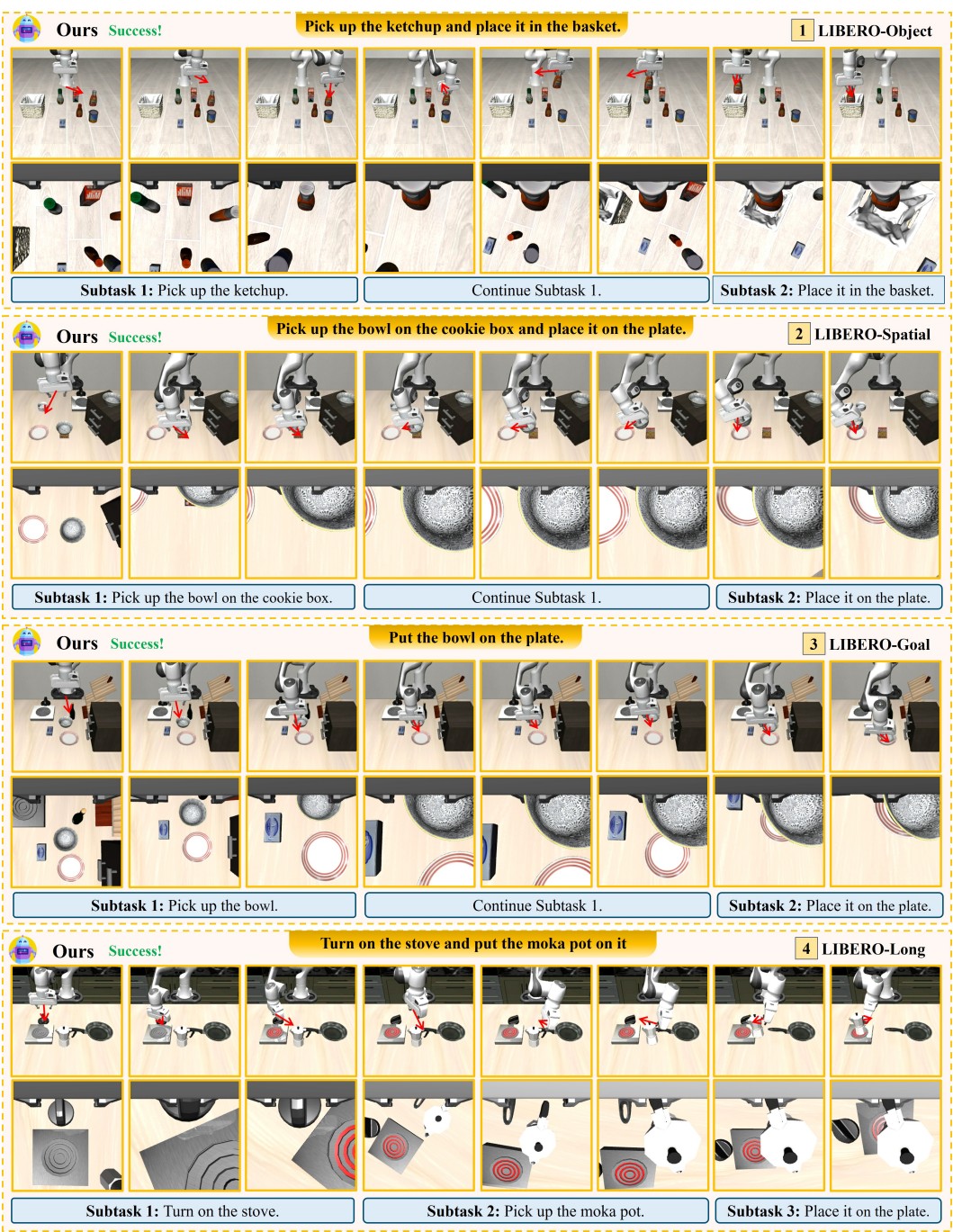

Figure 5: **Task domains used in our evaluation.** Across four domains, we evaluate our Agentic Robot on the LIBERO benchmark, including LIBERO-Object, LIBERO-Spatial, LIBERO-Goal, and LIBERO-Long.

## E    PSEUDO CODE OF AGENTIC ROBOT

---

**Algorithm 1** Agentic Robot Control Loop for Long-Horizon Tasks

---

1: **Input:** Task instruction $T$, initial observation $I_0$
2: **Output:** Task Success or Failure
3: $\{t_1, \ldots, t_N\} \leftarrow P(T, I_0)$        ▷ Planner: decompose high-level task $T$ into $N$ subgoals
4: $i \leftarrow 1; s \leftarrow 0; r \leftarrow 0$
5: **while** $i \leq N$ **do**
6:    $O_t \leftarrow \{I_t^r, I_t^w\}$             ▷ Multimodal Perception
7:    $\mathbf{a}_t \leftarrow \pi_{\text{exec}}(t_i, O_t)$         ▷ Reactive Execution for current subgoal
8:    $s \leftarrow s + 1$              ▷ Increment step counter
9:    **if** $s > S_{\max}$ **then**
10:      **return** Failure           ▷ Exceeded step limit
11:    **end if**
12:    **if** $s \bmod F = 0$ **then**        ▷ Perform verification every $F$ frames
13:      $done \leftarrow \pi_{\text{ver}}(\mathcal{B}_t, t_i)$      ▷ Primary verification: subgoal completion
14:      **if** $done$ **then**
15:        $i \leftarrow i + 1; r \leftarrow 0$    ▷ Success: move to next subgoal, reset recovery counter
16:      **else**
17:        $stuck \leftarrow \pi_{\text{diag}}(\mathcal{B}_t)$      ▷ Secondary check: is the arm stuck?
18:        **if** $stuck$ **then**
19:          $\mathbf{a}_{t+1} \leftarrow \pi_{\text{rec}}(stuck, O_{t+1})$    ▷ Trigger recovery (e.g., lift gripper)
20:          $r \leftarrow r + 1$
21:          **if** $r > R_{\max}$ **then**
22:            **return** Failure        ▷ Recovery limit exceeded
23:          **end if**
24:        **end if**
25:      **end if**
26:    **end if**
27: **end while**
28: **return** Success               ▷ All subgoals completed

---

## F    PSEUDO CODE FOR PERCEPTION-BASED VERIFIER

---

**Algorithm 2** VLM-Based Subgoal Verification

---

**Require:** Verifier model $\pi_{\text{ver}}$, processor, image buffer $B_t = \{(I_{t-k}^r, I_{t-k}^w)\}_{k=0}^{K-1}$, current subgoal $t_i$
**Ensure:** Binary result $y_t \in \{\text{YES}, \text{NO}\}$ indicating subgoal completion
1: Parse $t_i$ to extract verb $v$, object $o$, and location $l$
2: Construct prompt for subgoal completion according to $v$, $o$, and $l$
3: Initialize message buffer $\mathcal{M} \leftarrow [\,]$
4: **for** each image pair $(I_k^r, I_k^w)$ in $B_t$ **do**
5:    Append labeled image pair to $\mathcal{M}$
6: **end for**
7: Append constructed prompt to $\mathcal{M}$
8: Format $\mathcal{M}$ using processor template
9: (text, images) $\leftarrow$ process_vision_info($\mathcal{M}$)
10: Tokenize inputs and forward to $\pi_{\text{ver}}$
11: Decode response $r_t$
12: **if** $r_t$ starts with "Yes" **then**
13:    **return** YES              ▷ Subgoal completed
14: **else**
15:    **return** NO               ▷ Subgoal incomplete
16: **end if**

---

## G  PROMPT OF LRM FOR ZERO SHOT TASK DEVISION

---

**Prompt of LRM for Zero Shot Task Devision**

**You are a planning assistant for a fixed robotic arm. Your goal is to break down a high-level task into a sequence of \*\*essential high-level commands\*\*, suitable for a capable Vision-Language-Action (VLA) model to execute directly.**

*Output Format:*
Generate a numbered list of commands. Each command should represent a significant action achieving a clear sub-goal. Stick to the allowed high-level actions.

Example Plan Format (Use \*\*exactly\*\* this level of granularity):
Plan for the robot arm:

*Instructions:*

> - Generate \*\*only\*\* high-level commands.
> - \*\*Allowed commands are strictly limited to:\*\*
>   - `pick up [object]`
>   - `place [object] in/on [location]`
>   - `open [object/container/drawer/etc.]`
>   - `close [object/container/drawer/etc.]`
>   - `turn on [device]`
>   - `turn off [device]`
> - Use the commands above \*\*only when necessary\*\* to achieve the goal. Most tasks will primarily use `pick up` and `place`.
> - \*\*Explicitly DO NOT include separate steps for:\*\*
>   - `locate` (Assume VLA finds the object as part of executing the command)
>   - `move to` or `move towards` (Assume the command includes necessary travel)
>   - `lift`, `lower`, `grasp`, `release`, `push`, `pull`, `rotate`, `adjust` (Assume high-level commands handle these internally)
> - \*\*Assume the VLA model handles all implicit actions:\*\*
>   - "pick up [object]" means: Find the object, navigate to it, grasp it securely, and lift it.
>   - "place [object] in [location]" means: Transport the object to the location, position it correctly, and release the grasp.
>   - "open/close [container]" means: Find the handle/seam, interact with it appropriately (pull, slide, lift) to change the container's state.
>   - "turn on/off [device]" means: Find the correct button/switch, interact with it to change the device's power state.
> - Use the descriptive names from the task description (e.g., "alphabet soup", "basket", "stove", "microwave", "bottom drawer").
> - Generate the minimal sequence of these high-level commands required to fulfill the Goal. Ensure the sequence logically achieves the task (e.g., you might need to `open` a drawer before `place`ing something inside it, even if 'open' isn't explicitly stated in the goal).

*Task:* {task_description}
*Output:*

---

## H  LONG-HORIZON MANIPULATION ANALYSIS

We analyze performance on LIBERO-Long, which features multi-step tasks with sequential subgoals. Table 3 reports subgoal-level and overall success rates across 10 manipulation scenarios. Detailed task descriptions are provided in Appendix D.

Table 3: Performance comparison between OpenVLA and Agentic Robot on LIBERO-Long tasks, showing success rates for individual subtasks and overall task success rate (SR).

| Model | Soup-Sauce | | | Cheese-Butter | | | Stove-Moka | | | Bowl-Drawer | | | Mug-Mug | | |
|---|---|---|---|---|---|---|---|---|---|---|---|---|---|---|---|
| | Soup | Sauce | SR | Cheese | Butter | SR | Stove | Moka | SR | Bowl | Drawer | SR | Mug | Mug | SR |
| OpenVLA | 0.72 | 0.52 | 0.46 | 0.86 | 0.64 | 0.64 | 0.88 | 0.64 | 0.64 | 0.44 | 0.32 | 0.32 | 0.60 | 0.58 | 0.44 |
| **Agentic Robot** | **0.83** | **0.67** | **0.67** | **0.88** | **0.78** | **0.78** | **0.88** | **0.71** | **0.71** | **0.72** | **0.56** | **0.56** | **0.71** | **0.63** | **0.63** |

| Model | Book-Caddy | | | Mug-Pudding | | | Soup-Cheese | | | Moka-Moka | | | Mug-Wave | | |
|---|---|---|---|---|---|---|---|---|---|---|---|---|---|---|---|
| | Book | Caddy | SR | Mug | Pudding | SR | Soup | Cheese | SR | Moka | Moka | SR | Mug | Wave | SR |
| OpenVLA | 0.98 | 0.82 | 0.82 | 0.64 | 0.54 | 0.54 | 0.72 | 0.64 | 0.60 | 0.58 | 0.22 | 0.22 | 0.54 | 0.50 | 0.46 |
| **Agentic Robot** | **0.98** | **0.84** | **0.84** | **0.65** | **0.60** | **0.60** | **0.83** | **0.64** | **0.64** | **0.64** | **0.17** | **0.17** | **0.66** | **0.72** | **0.58** |

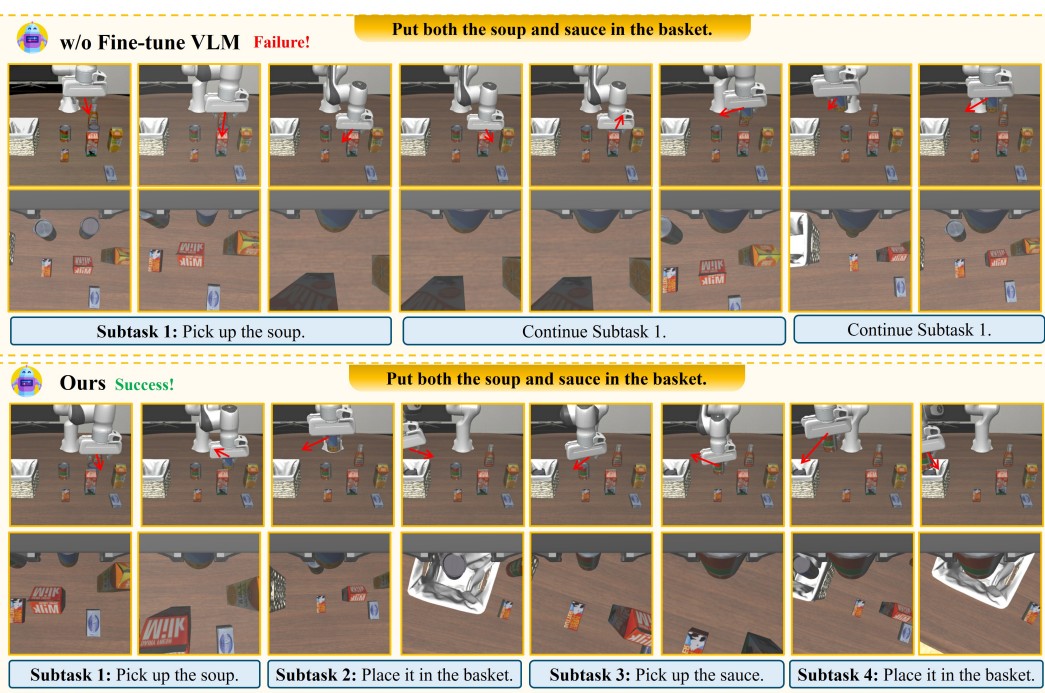

Figure 6: Comparison between **w/o fine-tuned VLM** and **Ours (w/ fine-tuned VLM)** on the task: "Put both the soup and the sauce in the basket." **Top:** Without fine-tuning, the VLM verifier fails to detect subtask completion, causing the robot to repeatedly attempt Subtask 1 (picking up the soup) until it ultimately fails. **Bottom:** With a fine-tuned VLM, each subtask is verified successfully before transitioning to the next, enabling sequential and successful execution of all subtasks.

**Performance gains.** The Agentic Robot consistently outperforms OpenVLA, with an average improvement of 12.1% across all tasks. The most significant gains are observed in challenging scenarios that have lower baseline success rates, such as the following: in the Bowl-Drawer task, there is a 24% improvement; in the Mug-Mug task, a 19% improvement; and in the Soup-Sauce task, a 21% improvement. These results confirm the effectiveness of our approach in reducing cascading failures.

**Error prevention.** OpenVLA frequently propagates errors through subtasks by continuing even when execution is incomplete. In contrast, Agentic Robot employs VLM-based verification, which ensures that progression cannot occur until each subgoal is confirmed as complete. This mechanism not only halts premature progression but also avoids redundant attempts on already completed subtasks, enabling efficient and orderly subgoal switching. This approach accounts for the improvements observed in spatially complex tasks like Stove-Moka, which saw a 7% increase in performance, highlighting the importance of robust checkpoint verification.

**Limitations.** In the Moka-Moka task, the Agentic Robot demonstrates decreased performance in one subgoal (17% vs. 22%), highlighting a limitation in managing scenes that require fine-grained coordination for the placement of symmetric objects. A qualitative analysis shows that while the first Moka Pot is placed correctly, the system often centers it on the stove surface. This positioning leaves insufficient space for the second Moka Pot, resulting in a placement failure despite having a correct high-level plan. This situation underscores the current weakness of the Agentic Robot in anticipating spatial constraints and resolving conflicts between similar subgoals that involve identical object types. Future work may address this issue by implementing temporal structure modeling, spatial intent prediction, or memory-aware policies that explicitly consider previous placements during planning and execution.

## I  VERIFICATION FREQUENCY ANALYSIS

We investigate how verification frequency affects both performance and computational efficiency by evaluating Agentic Robot with verification every 10, 20, or 50 steps. Fig. 7 presents the results.

**Success rate sensitivity.** All verification frequencies yield similar success rates for the Spatial, Object, and Goal suites, indicating that these shorter tasks are resilient to verification sparsity due to their limited subgoal durations and lower risk of error propagation. In contrast, LIBERO-Long shows considerable sensitivity to verification frequency. When verification is reduced to every 50 steps, the success rate decreases by 6 percentage points (from 61.8% to 55.8%). Providing verification every 10 steps does not offer any additional benefits beyond the 20-step interval. This suggests that long-horizon tasks require frequent validation to prevent error accumulation, but the benefits diminish beyond an optimal verification frequency.

**Efficiency-Performance trade-off.** Verification frequency substantially impacts execution time. High-frequency verification (every 10 steps) increases episode duration due to VLM inference overhead, particularly in LIBERO-Long, where the runtime difference between 10-step and 50-step intervals exceeds 15 seconds per episode.

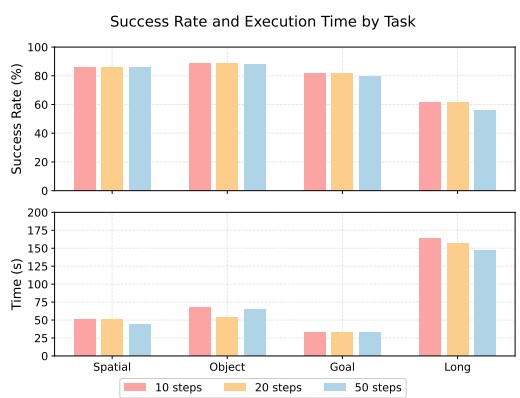

Figure 7: Effect of verification frequency on performance across LIBERO task suites. Bars compare three settings where the VLM verifier is invoked every 10, 20, or 50 steps during execution.

**Optimal configuration.** Our analysis identifies the 20-step interval as optimal: it maintains peak success rates while reducing computational overhead. We adopt this as our default configuration. These findings suggest that while frequent verification is critical for long-horizon robustness, it introduces unnecessary computational cost for simpler tasks. Future work could explore adaptive verification schemes that dynamically adjust frequency based on task complexity, execution uncertainty, or environmental dynamics.

**Adaptive verification scheduling.** Currently, verification occurs at fixed intervals (every 20 frames), regardless of task complexity, execution speed, or object dynamics. Although effective in our evaluations, this heuristic approach is likely suboptimal for computational efficiency. We propose to explore adaptive verification strategies that leverage confidence-aware scheduling based on motion intensity, subgoal typology, or the LLM's uncertainty quantification. Such approaches would optimize computational resource allocation while maintaining task safety and correctness guarantees.

Table 4: Additional LIBERO task performance results. Success rates (SR) across LIBERO benchmark task suites and results include another three baselines ($\pi_0$, $\pi_0$-FAST, and OpenVLA-OFT). Baseline results are from the original papers. Bold and underlined values indicate best and second-best performance.

| Method | Spatial SR (%) | Object SR (%) | Goal SR (%) | Long SR (%) | Average SR (%) |
|---|---|---|---|---|---|
| Diffusion Policy (Chi et al., 2023) | 78.3 | 92.5 | 68.3 | 50.5 | 72.4 |
| Octo-Base (FT) (Team et al., 2024) | 78.9 | 85.7 | 84.6 | 51.1 | 75.1 |
| OpenVLA (FT) (Kim et al., 2024) | 84.7 | 88.4 | 79.2 | 53.7 | 76.5 |
| TraceVLA (FT) (Zheng et al., 2024) | 84.6 | 85.2 | 75.1 | 54.1 | 74.8 |
| SpatialVLA (FT) (Qu et al., 2025) | 88.2 | 89.9 | 78.6 | 55.5 | 78.1 |
| $\pi_0$-FAST (FT) (Pertsch et al., 2025) | 96.4 | 96.8 | 88.6 | 60.2 | 85.5 |
| $\pi_0$ (FT) (Black et al., 2024) | 96.8 | 98.8 | 95.8 | 85.2 | 94.2 |
| OpenVLA-OFT (FT) (Kim et al., 2025) | **97.6** | 98.4 | **97.9** | **94.5** | **97.1** |
| **Agentic Robot Spatial** (SpatialVLA (FT) + SAP) | 88.8 | 90.1 | 80.2 | 62.3 | 80.4 |
| **Agentic Robot 2** ($\pi_0$ (FT) + SAP) (Ours) | 96.7 | **98.9** | 96.0 | 90.1 | 95.4 |

## J  ADDITIONAL EXPERIMENTS ON LIBERO BENCHMARK

Table 4 reports extended LIBERO task performance with additional baselines, including $\pi_0$, $\pi_0$-FAST, and OpenVLA-OFT, the latter representing the current state-of-the-art. As expected, OpenVLA-OFT achieves the highest overall success rate (97.1%), establishing a strong upper bound. Nevertheless, our framework demonstrates consistent advantages on long-horizon tasks. Specifically, when augmenting SpatialVLA with the SAP protocol (*Agentic Robot Spatial*), performance improves from 55.5% to 62.3% on LIBERO-Long, a gain of +6.8 absolute points. This validates that verifier-based termination and recovery mechanisms are especially beneficial for extended manipulation sequences, where compounding errors are most detrimental.

When integrated with the stronger $\pi_0$ backbone (*Agentic Robot 2*), our approach achieves 90.1% on LIBERO-Long, a notable improvement over $\pi_0$ (85.2%) and $\pi_0$-FAST (60.2%). Importantly, this setting reaches an overall average of 95.4%, placing it second only to OpenVLA-OFT while retaining clear advantages on challenging long-horizon categories. These results highlight the complementary role of SAP: while the backbone determines single-pass proficiency, the closed-loop verification-feedback design improves robustness and efficiency in multi-stage manipulation.

Throughout these experiments, SAP is instantiated with GPT-4o as the planner and a LoRA-fine-tuned Qwen2.5-VL verifier, consistent with our core framework description. Crucially, the design remains modular: alternative LRMs or VLMs can be substituted without changing the coordination protocol. This modularity paves the way for further performance gains by pairing SAP with more advanced reasoning or verification models as they become available.

Together, these findings reinforce our central claim: SAP primarily strengthens reliability in long-horizon tasks, without compromising the performance of competitive backbone executors. Although OpenVLA-OFT remains the strongest one-pass model, our approach delivers modular improvements that generalize across different baselines and provide systematic error mitigation in extended task horizons.

