# OpenReview forum: "Agentic Robot: A Brain-Inspired Framework for Vision-Language-Action Models in Embodied Agents"
_ICLR.cc/2026/Conference — ICLR 2026 Conference Withdrawn Submission_

### Official Review · Reviewer_SuHz · 2025-10-21

**Soundness:** 2
**Presentation:** 3
**Contribution:** 1
**Rating:** 4
**Confidence:** 4

**Summary:**

The paper introduces Agentic Robot, a brain-inspired embodied manipulation framework integrating planning, execution, and verification via a Standardized Action Procedure (SAP). The approach outperforms prior methods on the LIBERO benchmark (79.6% success), with ablation studies confirming the benefits of subgoal decomposition and visual verification.

**Strengths:**

1. The paper features a clear modular design: the planner–executor–verifier loop is explicitly defined and highly interpretable, drawing a meaningful analogy to biological and cognitive systems.
2. The paper is clearly organized and well written, making it easy to follow the methodology and key ideas.

**Weaknesses:**

1. The novelty of the work is limited. Each module is built on existing systems, and the main contribution lies in their integration rather than model-level innovation. Much of the work focuses on prompt engineering and system composition. The overall pipeline is very similar to many existing agent-based frameworks, showing little distinction from prior work [1,2,3].
2. A key limitation is the lack of real-world validation. All experiments are performed in simulation, while physical robot deployment is only mentioned conceptually without empirical demonstration.
3. The evaluation is restricted to LIBERO, and generalization to other domains or unseen settings remains untested.

[1] Wang, Zihao, et al. "Describe, explain, plan and select: interactive planning with large language models enables open-world multi-task agents." Proceedings of the 37th International Conference on Neural Information Processing Systems. 2023.

[2] Yao, Shunyu, et al. "React: Synergizing reasoning and acting in language models." The eleventh international conference on learning representations. 2022.

[3] Du, Yuqing, et al. "Vision-Language Models as Success Detectors." Conference on Lifelong Learning Agents. PMLR, 2023.

**Questions:**

1. See weaknesses above.
2. In addition to its effect on overall task success rate, it would be helpful to include a quantitative evaluation of the verifier’s accuracy in assessing whether a task has been successfully completed.
3. The recovery mechanism is rather simplistic, consisting mainly of lifting and retrying. According to the results, this leads to a 59.7% success rate, representing only a 1.9% change—an effect that is marginal and inconsistent with the claimed advantage of the proposed recovery strategy.

---

### Official Review · Reviewer_PSLS · 2025-10-21

**Soundness:** 3
**Presentation:** 3
**Contribution:** 3
**Rating:** 2
**Confidence:** 3

**Summary:**

The paper proposes Agentic Robot, a brain-inspired control framework for long-horizon manipulation built around a Standardized Action Procedure (SAP) that coordinates three modules: an LRM-based planner that decomposes tasks into subgoals, a VLA executor, and a VLM verifier for periodic success checking and recovery. Experiments on LIBERO report an average success rate of 79.6%, with the largest gains on LIBERO-Long

**Strengths:**

- Clear modularity and interpretability. SAP defines explicit interfaces and a closed perception–planning–execution–verification loop, making component roles and information flow easy to reason about.
- Verification as a control signal. The periodic VLM verifier enables early error detection, retries, and subgoal-level termination, which is well-motivated for long horizons.

**Weaknesses:**

- Limited technical novelty. The core contribution is a coordination protocol that combines existing ingredients (LLM/LRM planner, VLA executor, VLM verifier) rather than introducing new learning algorithms or model architectures; much of the lift comes from structuring the pipeline.
- Baselines skew toward end-to-end VLAs. Comparisons largely pit SAP against executor-only policies; the paper lacks head-to-head evaluations versus hierarchical planner + skill/executor systems (e.g., code-as-policies / voxel-value-map styles or recent cognition–execution bridges). This makes it hard to attribute gains specifically to SAP versus generic hierarchical decomposition.
- Magnitude of improvements is uneven. Average performance is competitive but only modestly above strong VLAs on non-long suites; the main benefits are concentrated in LIBERO-Long. A deeper analysis of where SAP helps (failure modes, horizon length) would strengthen the case.
- No real-robot validation. Results are simulation-only; the paper itself acknowledges challenges for physical deployment and domain shift, leaving sim-to-real efficacy untested.

**Questions:**

- Comparative scope. Can you include baselines that combine a high-level planner with a low-level executor/skills (not just end-to-end VLAs), to isolate the effect of SAP’s verification/coordination versus generic hierarchy? (Even a re-implementation with simple rule-based termination would be informative.)
- Verifier reliability & overhead. What are the verifier’s false-positive/negative rates, and how sensitive is performance to the verification frequency? Any ablation on adaptive scheduling or computational footprint under different intervals?

---

### Official Review · Reviewer_hZvB · 2025-10-28

**Soundness:** 2
**Presentation:** 3
**Contribution:** 2
**Rating:** 2
**Confidence:** 4

**Summary:**

This paper proposes Agentic Robot, a brain-inspired framework for long-horizon robotic manipulation that coordinates planning, execution, and verification via a Standardized Action Procedure (SAP). On the LIBERO benchmark, Agentic Robot attains a 79.6% average success rate, outperforming SpatialVLA by 6.1% and OpenVLA by 7.4% on long-horizon tasks. The approach improves both performance and interpretability by making subgoal-level monitoring and recovery explicit.

**Strengths:**

- Originality: The paper introduces a clearly defined coordination protocol—Standardized Action Procedure (SAP)—that operationalizes a brain-inspired perception–planning–execution–verification loop for embodied agents. Unlike prior sequential planners or end-to-end VLAs, SAP formalizes subgoal-level verification and recovery as first-class components, yielding a modular and enforceable protocol that goes beyond ad-hoc prompting or implicit success checks.

- Quality: The framework is carefully engineered with three specialized, interoperable modules (LRM planner, VLA executor, VLM verifier).  The temporal verifier design (sliding buffer, two-stage assessment/diagnosis, recovery) is well-motivated and empirically supported, showing robust gains on long-horizon tasks.

**Weaknesses:**

- The evaluation is overly simple and limited: it relies solely on the LIBERO benchmark and includes no real-world experiments.
- It omits comparisons with stronger open-source baselines, such as pi0.5.

**Questions:**

See Weaknesses.

---

### Official Review · Reviewer_fqQc · 2025-10-29

**Soundness:** 2
**Presentation:** 3
**Contribution:** 3
**Rating:** 6
**Confidence:** 3

**Summary:**

This paper introduces Agentic Robot, a brain-inspired framework designed to improve long-horizon robotic manipulation through structured coordination among specialized components. The central idea is the Standardized Action Procedure (SAP), a protocol inspired by Standardized Operating Procedures (SOPs) used in human organizations, which formalizes the workflow across planning, execution, and verification phases. The framework integrates three key modules: a large reasoning model for task decomposition, a vision-language-action executor for continuous control, and a temporal verifier for self-monitoring and error recovery. This design enables dynamic self-verification and robust subgoal transitions without external supervision. Experiments on the LIBERO benchmark show that Agentic Robot achieves competitive results, attaining a 79.6% average success rate and outperforming prior systems such as SpatialVLA and OpenVLA, particularly on long-horizon tasks. The findings suggest that structured inter-component coordination can enhance both the reliability and interpretability of autonomous robotic systems.

**Strengths:**

1. Overall, this is a well-written paper with good motivation and experiment validation.

2. A lot of details are shown in the appendix.

**Weaknesses:**

1. No real-world deployment experiments.

2. More datasets and baselines should be included like VLABench, Cot-VLA, COA-VLA, pi 0.5, Groot, etc.

https://openaccess.thecvf.com/content/ICCV2025/papers/Li_CoA-VLA_Improving_Vision-Language-Action_Models_via_Visual-Text_Chain-of-Affordance_ICCV_2025_paper.pdf

https://arxiv.org/pdf/2501.15830

https://openaccess.thecvf.com/content/CVPR2025/papers/Zhao_CoT-VLA_Visual_Chain-of-Thought_Reasoning_for_Vision-Language-Action_Models_CVPR_2025_paper.pdf

https://arxiv.org/pdf/2508.07917

https://arxiv.org/pdf/2503.14734

3. Some of the related work should be discussed. The subtask planning and self-reflection for long-term tasks are well explored and should be discussed.

https://arxiv.org/abs/2411.18279

https://arxiv.org/abs/2311.15649

https://arxiv.org/abs/2403.03186

https://arxiv.org/abs/2508.10146

https://arxiv.org/abs/2504.16054

**Questions:**

N/A

---

### Note · Authors · 2025-11-15

I have read and agree with the venue's withdrawal policy on behalf of myself and my co-authors.